# A Causal Viewpoint on Motor-Imagery Brainwave Decoding

**Konstantinos Barmpas**
Imperial College London & Cogitat
`konstantinos.barmpas16@imperial.ac.uk`

**Yannis Panagakis**
National and Kapodistrian University of Athens & Cogitat

**Dimitrios A. Adamos**
Imperial College London & Cogitat

**Nikolaos Laskaris**
Aristotle University of Thessaloniki & Cogitat

**Stefanos Zafeiriou**
Imperial College London & Cogitat

## Abstract

In this work, we employ causal reasoning to breakdown and analyze important challenges of the decoding of Motor-Imagery (MI) electroencephalography (EEG) signals. Furthermore, we present a framework consisting of dynamic convolutions, that address one of the issues that arises through this causal investigation, namely the subject distribution shift (or inter-subject variability). Using a publicly available MI dataset, we demonstrate increased cross-subject performance in two different MI tasks for four well-established deep architectures.

## 1 Introduction

Brain-Computer Interface (BCI) technology primarily aspires to provide a direct neural communication and control between individuals (subjects) and computers. This is feasible by analyzing brainwaves captured by EEG signal recordings using signal processing and Machine Learning (ML) techniques. One of the first and most popular BCI paradigms is Motor-imagery (MI). MI-BCIs are based on a cognitive process, by which a subject mentally simulates a motor action, for example the movement of a hand or foot, without actually executing it (Decety & Ingvar (1990)). Developing MI-BCI systems mainly relies on robust decoding of a subject's motor intentions from the recorded EEG signals, under the prior assumption that these signals encode that relevant information, and are mainly used for movement rehabilitation purposes (e.g. Mane et al. (2020), Robinson et al. (2021), (Sebastián-Romagosa et al. (2020)).

In recent years, Deep Learning (DL) techniques - and most specifically Convolutional Neural Networks (CNNs) - have largely alleviated the need for manual feature extraction, achieving state-of-the-art performance in various areas, most notably Computer Vision (Chai et al. (2021)). Due to their massive progress, CNN-based feature extractors have been introduced in various paradigms in the field of BCIs (e.g. Antoniades et al. (2016), Rezaeitabar & Halici (2017), Längkvist et al. (2012), Wulsin et al. (2011)), in an effort to become generic EEG signal processing tools. One of the core challenges that a BCI - or more generally the decoding of EEG signals - faces is to cope with changes in data distributions across different subjects. Each individual has a unique brain anatomy and functionality that makes the discovery and exploitation of shared invariant features extremely difficult. Therefore, modern DL-based BCIs tend to fail to generalize well in unseen subjects due to this type of data distribution shift.

Causal reasoning provides tools to breakdown and analyze important aspects of a BCI task, identify and possibly resolve some of these challenges by employing appropriate ML strategies. The methodical breakdown of a BCI task and the identification of the causal relationships between the various variables of interest take into account the expert's knowledge of the involved biological and neurophysiological processes and can be of vital importance when designing and building ML-based models in the field of BCI. In this work, we focus mainly on MI-BCI systems, and inspired by the

work of (Schölkopf et al. (2021)), we analyze the task of MI EEG signal classification through the lens of causal reasoning. Motivated by this causal analysis, we introduce a framework based on dynamic convolutions that tackles the identified problem of subject distribution shift.

## 2 CAUSALITY IN MOTOR-IMAGERY BRAINWAVE DECODING

### 2.1 TASK DEFINITION

In a Motor-Imagery (MI) classification problem, we want to accurately predict the mentally performed task from a recorded EEG signal. Mathematically, given an input EEG signal $X$, we train a statistical model to predict the correct MI task $Y$, which can be the imagery movement of a hand or foot. In essence, this statistical model tries to estimate the conditional probability $P(Y|X)$ using an appropriate objective function. Causal reasoning is the analysis of a task / problem in terms of cause-effect relationships between the different variables of interest: if a variable A is a direct cause of variable B, we express it as $A \rightarrow B$ (A causes B or B is the effect of A).

In ML-based tasks, given the input $X$ and the prediction target $Y$, we can establish that the task to estimate $P(Y|X)$ can be either (Castro et al. (2020)):

- Causal: when $X \rightarrow Y$, prediction effect from cause
- Anti-causal: when $Y \rightarrow X$, prediction cause from effect

Using the above basis, we can define an MI EEG classification task as an anti-causal problem, since the true MI intention (observed with the MI label $Y$) can be considered the cause of the recorded EEG signal $X$. Additionally, inspired by (Castro et al. (2020)), we can consider $X$ as a sequence of imperfect observed measurements (in sensor-space) of the true unobserved brain activations $Z$ within, mainly, the cortical areas responsible for the sensorimotor rhythms, i.e. $Z \rightarrow X$. Therefore, using a causal diagram, an MI EEG classification task can be described as:

$$X \leftarrow Z \leftarrow Y \tag{1}$$

As a consequence of the above anti-causal definition and causal diagram, we can explore the problem of MI EEG classification through the following causal factorization:

$$P(X, Y, Z) = P(X|Z)P(Z|Y)P(Y) \tag{2}$$

### 2.2 TASK CHALLENGES

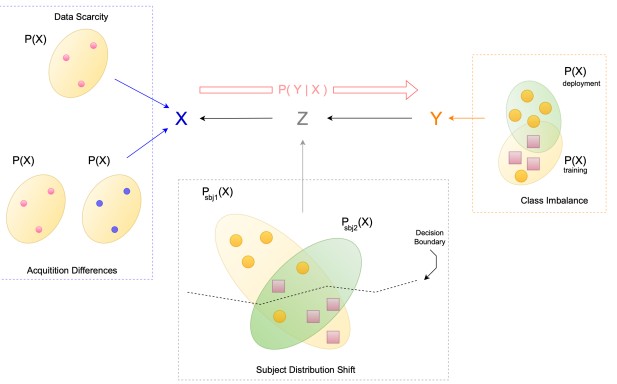

Figure 1: Key challenges in machine learning for a MI EEG classification task. X represents input EEG signals, Y the associated MI labels. Big circles and rectangles represent EEG signals of different labels. Dots represent data points of any label and their color represent different EEG acquisition devices.

Through this causal breakdown, we can categorize the major challenges associated with Motor-Imagery (MI) EEG classification tasks into three main categories. Challenges related with the:

1. **Training EEG signals.** One of the renowned challenges in motor-imagery classification problem - as in any medical-related machine learning problem - is the scarcity of labelled data due to the lengthy acquisition process. Subjects are required to spend hours in a laboratory facility performing subsequent motor-imagery tasks. This process has been reported to cause fatigue and discomfort, even when devices with dry electrodes are utilized. To make things worse, due to the wide variety of available EEG recorders in the market, the data acquisition can be undertaken with various devices which have completely different specifications (e.g. number of electrodes, sampling frequency to name just a few), making the combination of publically available EEG datasets extremely difficult.

2. **Anatomical differences of subjects.** Each subject has a unique brain anatomy and functionality that results in polymorphous neural activity patterns when appeared in the surface observed EEG signal. When designing a generic ML-based MI-BCI, researchers need to take this inter-subject variability (subject distribution shift) into account.

3. **Class Imbalance.** Class imbalances can arise between the training and the deployment set of a MI-BCI. It is necessary for the training set to be as closely balanced to the deployment set as possible when training machine learning models.

## 3 PROPOSED FRAMEWORK

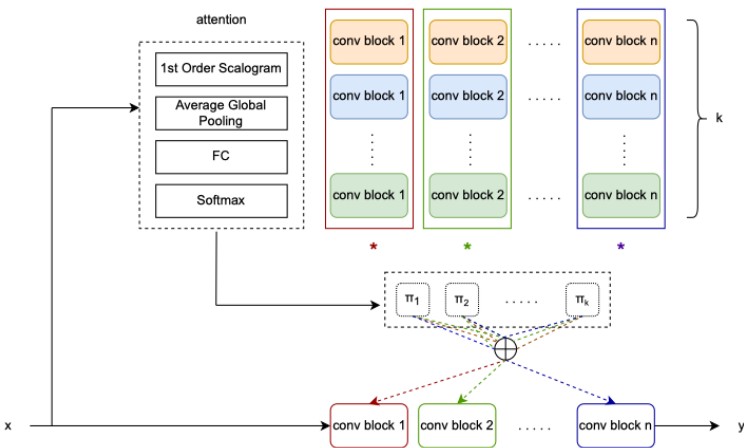

Figure 2: Dynamic convolution framework for BCI architectures. X represents input EEG signals, Y the associated MI labels. K different subjects in the training set are represented by different colors in the convolutional blocks. Colored rectangles and arrows (namely green, red and dark blue) demonstrate the different blocks that are taken into account when computing the final convolutional blocks for the MI classification task.

In this work, we mainly focus on the challenge of subject distribution shift (or inter-subject variability). As described in the next section, we will use a publicly available MI dataset - which contains a large number of different subjects, is class balanced, has relatively enough trials per subject and all trials come from a single EEG recorder, essentially solving all the above identified challenges but the subject distribution shift. In terms of the causal factorization 2, the problem of inter-subject variability can be seen as a distribution shift $S$ where:

$$P(X, Y, Z) = P(X|Z)P_{\mathbf{S}}(Z|Y)P(Y) \tag{3}$$

Here, we propose a framework that tackles the problem of subject distribution shift (or inter-subject variability) and can be applied to any established CNN-based MI-BCI architecture, resulting in a statistically significant performance increase.

Our framework is based on the concept of dynamic convolutions as originally introduced in (Chen et al. (2020)). In their work, they present dynamic convolution layer as an effective way to increase model's complexity without increasing the network's depth and / or width, since each convolutional layer is computed by dynamically mixing multiple parallel learnt convolutional kernels based on an input-dependent attention vector. Inspired by this concept, we utilize dynamic convolutions in the domain of MI brainwave decoding. Instead of having a BCI architecture that tries to discover a common latent space for all k subjects in the training set, we use k parallel trainable convolutional kernels (corresponding to the k available training subjects) for each convolutional block of a CNN-based BCI network. Using a subject attention network that learns to distinguish between the available individuals, we decouple the subjects and essentially train simultaneously k parallel personalized models of the same BCI architecture, as illustrated in Figure 2.

Our subject attention network consists of the first order wavelet scalogram of the input EEG signal $X$ followed by a global average pooling across time and frequency. Mathematically, let $\mathbf{x}(t) \in \mathbb{R}^T$ denote a one-dimensional input EEG signal, where T is the number of initial EEG time points, and $\boldsymbol{\psi}_\lambda(t)$ be a wavelet. The 1st order scalogram is defined as $\mathbf{X}(\lambda, t) = |\mathbf{x}(t) * \boldsymbol{\psi}_\lambda(t)|$. To perform this operation, the raw input signal from each EEG channel is convolved with a wavelet kernel with size $(1, W) = (1, \frac{F_s}{2})$ where $F_s$ is the sampling frequency. This wavelet kernel follows the real Gabor wavelet format:

$$\boldsymbol{\psi}_\lambda(t) = \frac{1}{\sqrt{2\pi}\sigma} e^{-\frac{t^2}{2\sigma^2}} cos(2\pi\lambda t) \tag{4}$$

with $t \in [-\frac{W}{2}, ..., \frac{W}{2}]$ and $\frac{1}{\sigma}$ denotes the bandwidth and $\lambda$ the normalized frequency of the Gabor wavelet and these two properties are the only trainable parameters of this layer. During training, $\lambda$ is restricted ($\lambda \in [0, \frac{1}{2}]$) to satisfy the Nyquist theorem.

The proposed framework takes the EEG signal $X$ as input and tries to learn both the correct MI task $Y$ (estimate the conditional probability $P(Y|X)$) as well as the correct subject id $\pi$ (estimate the conditional probability $P(\pi|X)$). The subject attention network and the k parallel convolutional kernels are trained simultaneously using the following loss function:

$$Loss = (1 - acc) * \ell_{Attention} + acc * \ell_{MI} \tag{5}$$

where $acc$ is the training accuracy of the subject attention network and $\ell$ denotes the cross-entropy function ($\ell_{Attention}$ for the subject attention network and $\ell_{MI}$ for the MI classification task). This loss function effectively enforces first the training of the subject attention network and, as the attention's accuracy increases, it switches its focus to train the parallel convolutional kernels for the different MI tasks. As also suggested in (Chen et al. (2020)), since softmax does not work well due to its near one-hot output, we use a large temperature of 30 in the softmax of the attention network during training in order to flatten the framework's attention, allow a broader gradient backpropagation and effectively assist in the attention network's training in the early epochs. On the other hand, during evaluation we use the hard softmax (temperature of 1).

During inference, when an input EEG signal from a new unseen subject $S_x$ is processed, it passes firstly through the attention network and the subject attention vector $\boldsymbol{\pi}$ is computed. Empirically, it was shown that this vector is quite sparse, and if it was used during inference, only a handful of parallel convolutional kernels would be utilized during the kernel mixing. Instead, we would ideally like to use knowledge from all k individuals and "shift" the attention more to the most relevant subjects. To accomplish that, we compute what we call the "uniformly attended vector" **A\***. If there was no attention network, the k parallel convolutional kernel would be mixed with a uniform factor $A_i = \frac{1}{k}$. To compute the "uniformly attended vector", the uniform attention vector **A** is combined with the subject attention vector $\boldsymbol{\pi}$ and the result is passed through a softmax activation to flatten the attention across all subjects - while maintaining the focus on the most relevant ones. Mathematically, this operation can be described as:

$$\underbrace{\begin{pmatrix} A_1 \\ A_2 \\ ... \\ A_k \end{pmatrix}}_{\mathbf{A}} + \underbrace{\begin{pmatrix} \pi_1 \\ \pi_2 \\ ... \\ \pi_k \end{pmatrix}}_{\boldsymbol{\pi}} \xrightarrow{\sigma} \underbrace{\begin{pmatrix} A_1^* \\ A_2^* \\ ... \\ A_k^* \end{pmatrix}}_{\mathbf{A^*}} \tag{6}$$

where $\sigma$ denotes the softmax operation, $\mathbf{A}$ the uniform attention vector with $A_i = \frac{1}{k}$, $\boldsymbol{\pi}$ the subject attention vector with $\sum_i \pi_i = 1$ and $\mathbf{A^*}$ the "uniformly attended" vector where $\sum_i A_i^* = 1$.

In other words, using the causal factorization 3, our proposed framework tries to to estimate the $P_{S_x}(Z|Y)$ of a new unseeen subject $S_x$ as the linear combination of k learned conditional probabilities. More specifically:

$$P_{S_x}(Z|Y) = A_1^* \times P_{S_1}(Z|Y) + A_2^* \times P_{S_2}(Z|Y) + ... + A_k^* \times P_{S_k}(Z|Y) \tag{7}$$

## 4 EXPERIMENTS

We tested our proposed framework in four well-established BCI architectures, namely DeepConvNet (Schirrmeister et al. (2017)), ShallowConvNet (Schirrmeister et al. (2017)), EEGNet (Lawhern et al. (2018)) and EEG-Inception (Santamaría-Vázquez et al. (2020)) in two MI tasks formed based on the publically available MI dataset Physionet (Goldberger et al. (2000)). The one is a binary classification task (MI Left vs Right Hand) and the other is a 3-class classification problem (MI Left Hand / Right Hand / Feet). The original Physionet dataset includes brain recordings from 109 healthy participants, registered via 64 EEG sensors with a sampling frequency of 160 Hz, while performing a series of pseudo-randomized cue-triggered MI tasks. In our experiments, we first excluded data from 6 participants (subjects 88, 89, 92, 100, 104 and 106) due to differences in either the sampling frequency or duration of the performed tasks. We evaluated the performance of the normal networks and their equivalent dynamic networks in a leave-one-subject-out fashion. We trained the normal networks for 50 epochs with learning rate of 0.001 while their dynamic versions for 30 epochs (to avoid overfitting) - in the first 20 epochs with learning rate of 0.01, to assist the attention's Gabor filters to quickly adapt to the data, and 10 epochs with learning rate of 0.001 and frozen attention, to fine-tune to the MI task. In both cases, we use an Adam optimizer. Finally, we performed a paired t-test between the the normal and dynamic results.

| Network | Task | Normal Avg. Accuracy | Dynamic Avg. Accuracy | Paired t-Test |
|---|---|---|---|---|
| ShallowConvNet | MI Left vs Right Hand | 80.6% | **83.3%** | p-value ≈ **0.0055** |
| ShallowConvNet | MI Left / Right Hand / Feet | 66.3 % | **69.0%** | p-value ≈ **0.0043** |
| DeepConvNet | MI Left vs Right Hand | 82.5 % | **83.5%** | p-value ≈ 0.13 |
| DeepConvNet | MI Left / Right Hand / Feet | 67.1% | **71.3%** | p-value ≈ **0.0000328** |
| EEGNet | MI Left vs Right Hand | 79.5% | **80.2%** | p-value ≈ 0.26 |
| EEGNet | MI Left / Right Hand / Feet | 66.5% | **67.5%** | p-value ≈ 0.15 |
| EEGInception | MI Left vs Right Hand | 81.6% | **83.9%** | p-value ≈ **0.0068** |
| EEGInception | MI Left / Right Hand / Feet | 67.6% | **71.4%** | p-value ≈ **0.00025** |

Table 1: Performance of generic (trained and evaluated in a leave-one-subject-out fashion) models of MI-classification (Left / Right hand and Left / Right hand / Feet) tasks in Physionet for DeepConvNet, ShallowConvNet, EEGNet, EEG-Inception and their Dynamic equivalent networks

## 5 CONCLUSION

In this work, we analyze the task of MI EEG classification through the lens of causal reasoning. To the best of our knowledge, this is the first work that brings machine learning in conjunction with causal reasoning to the domain of EEG. Through this analysis, we identify and analyze some of the major challenges and we introduce a framework based on dynamic convolutions that tackles the problem of subject distribution shift (inter-subject variability). Our proposed framework demonstrates increased performance when applied to different BCI architectures. Although this constitutes early results in this research direction, we strongly believe that the proposed framework can have further benefits in MI-BCIs and, in future work, we plan to use it to tackle more, if not all, challenges detailedly described in this causal analysis.

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
