# OpenReview forum: "A CAUSAL VIEWPOINT ON MOTOR-IMAGERY BRAINWAVE DECODING"
_ICLR.cc/2022/Workshop/OSC — ICLR2022 OSC  Poster_

### Official Review · Reviewer_k95d · 2022-03-14
**Review: A CAUSAL VIEWPOINT ON MOTOR-IMAGERY BRAIN-WAVE DECODING**

**Rating:** 2
**Confidence:** 1

**Review:**

**Summary**: The authors present a method leveraging dynamic convolutions (Chen et al 2020) to handle distribution shift between individual test subjects’ EEG recordings for the purpose of motor-imagery decoding. They then demonstrate statistically significant classification accuracy improvements across a range of convolutional neural network architectures and tasks.

**Strong points**:
- The introduction of causal reasoning into EEG analysis is welcomed and I believe this is a good start in that direction.
- The empirical results are positive and statistical significance is reported.
- The idea of using dynamic convolutions for distribution shift is new to the best of my knowledge.

**Suggestions for improvement**:
- Many of the architecture choices appear specifically engineered beyond the baseline without ablation study or mention to prior work which justifies their use. For example, the purpose of the use of a wavelet scalogram for the subject attention network is not entirely clear. If this is common in the prior work, it would be helpful to note this. A second example is the ‘uniformly attended vector’ for which the stated motivation appears to run counter to the purpose of using the dynamic convolution in the first place – shouldn’t the subject attention vector naturally shift attention more to the most relevant subjects? If training is causing the vector to become overly sparse, it seems to make sense to change the training procedure or network architecture to prevent this rather than change the algorithm at inference time.
- The connection with causal reasoning is not as strong as it could be, or at least it is not entirely convincing from the paper’s current presentation. The motivation for equation (7) is clear, however the remaining (weakly-justified) components of Section 3 (as described above) significantly detract from the paper’s overall goal.

Overall, the motivation and direction of the paper are strong and I encourage the authors to explore this idea further. In its current state, my opinion is that this paper is a borderline case due to the above limitations (a minimal connection to causal reasoning, and relatively insufficient analysis of the complexities of the proposed method). However, I do believe there are valuable elements of this paper which put it marginally above the acceptance threshold.

---

### Official Review · Reviewer_Ryto · 2022-03-15
**comments for A CAUSAL VIEWPOINT ON MOTOR-IMAGERY BRAIN WAVE DECODING**

**Rating:** 2
**Confidence:** 3

**Review:**

This paper proposes a framework to tackle the MI EEG classification task and mainly focus on the challenge of subject distribution shift. Specifically, the core module of the framework is the concept of dynamic convolutions which contain parallel trainable convolutional kernels to increase model’s complexity without increasing the network’s depth or width.

However, it seems that the authours just apply this already proposed method to the task of MI EEG classification and the whole method lacks novelty. Besides,  causal reasoning is not well descrbed in the paper except Eq.(3) which is not enough for self-contained and Z is not stated in your pipeline (Fig. 2)

---

### Decision · Program_Chairs · 2022-03-24

Accept (Poster)